# Impact of Shape Irregularity in Medial Sphenoid Wing Meningiomas on Postoperative Cranial Nerve Functioning, Proliferation, and Progression-Free Survival

**DOI:** 10.3390/cancers15123096

**Published:** 2023-06-07

**Authors:** Johannes Wach, Johannes Naegeli, Martin Vychopen, Clemens Seidel, Alonso Barrantes-Freer, Ronny Grunert, Erdem Güresir, Felix Arlt

**Affiliations:** 1Department of Neurosurgery, University Hospital Leipzig, University of Leipzig, 04103 Leipzig, Germany; johannes.naegeli@medizin.uni-leipzig.de (J.N.); martin.vychopen@medizin.uni-leipzig.de (M.V.); ronny.grunert@medizin.uni-leipzig.de (R.G.); erdem.gueresir@medizin.uni-leipzig.de (E.G.); felix.arlt@medizin.uni-leipzig.de (F.A.); 2Department of Radiation Oncology, University Hospital Leipzig, University of Leipzig, 04103 Leipzig, Germany; clemens.seidel@medizin.uni-leipzig.de; 3Department of Neuropathology, University Hospital Leipzig, University of Leipzig, 04103 Leipzig, Germany; alonso.barrantes-freer@medizin.uni-leipzig.de; 4Fraunhofer Institute for Machine Tools and Forming Technology, Theodor-Koerner-Allee 6, 02763 Zittau, Germany

**Keywords:** medial sphenoid wing meningioma, MIB-1, oculomotor nerve palsy, progression, shape

## Abstract

**Simple Summary:**

The shape of meningiomas has been suggested as a potential indicator of the WHO grade. Medial sphenoid wing meningiomas are surgically challenging skull base tumors regarding the preservation of cranial nerve functioning. The present study investigates the impact of tumor shape on neuropathology, progression-free survival, and cranial nerve functioning. The present investigation shows that irregular shape is significantly associated with new postoperative cranial nerve deficits and a shorter time to tumor progression. From a pathological point of view, irregular shapes may result from areas with increased proliferative activity. A systematic review and pooled data analysis that included the present study revealed that an irregular shape is associated with a higher MIB-1 labeling index. Tumor shape should be considered in the preoperative surgical planning regarding the preservation of cranial nerve functioning. Further research is required to examine the molecular basis of irregular meningioma shape.

**Abstract:**

Medial sphenoid wing meningiomas (MSWM) are surgically challenging skull base tumors. Irregular tumor shapes are thought to be linked to histopathology. The present study aims to investigate the impact of tumor shape on postoperative functioning, progression-free survival, and neuropathology. This monocentric study included 74 patients who underwent surgery for primary sporadic MSWM (WHO grades 1 and 2) between 2010 and 2021. Furthermore, a systematic review of the literature regarding meningioma shape and the MIB-1 index was performed. Irregular MSWM shapes were identified in 31 patients (41.9%). Multivariable analysis revealed that irregular shape was associated with postoperative cranial nerve deficits (OR: 5.75, 95% CI: 1.15–28.63, *p* = 0.033). In multivariable Cox regression analysis, irregular MSWM shape was independently associated with tumor progression (HR:8.0, 95% CI: 1.04–62.10, *p* = 0.046). Multivariable regression analysis showed that irregular shape is independently associated with an increased MIB-1 index (OR: 7.59, 95% CI: 2.04–28.25, *p* = 0.003). A systematic review of the literature and pooled data analysis, including the present study, showed that irregularly shaped meningiomas had an increase of 1.98 (95% CI: 1.38–2.59, *p* < 0.001) in the MIB-1 index. Irregular MSWM shape is independently associated with an increased risk of postoperative cranial nerve deficits and a shortened time to tumor progression. Irregular MSWM shapes might be caused by highly proliferative tumors.

## 1. Introduction

Meningiomas originating from the lining of the central nervous system (CNS) are the most common primary intracranial neoplasm and account for 54.5% of all non-malignant tumors [1]. Meningiomas are thought to arise from derivates of the neural crest, an embryonic cell population known for its exceptional genetic and functional diversity, controlled by conserved gene regulatory systems [2,3]. The World Health Organization (WHO) classifies meningiomas based on mitotic activity, brain invasion, TERT promoter mutations, and other neuropathological characteristics [4]. Approximately 15–20% of all meningiomas grow along the sphenoid wing and are assessed based on their involvement of the sphenoid wing. These tumors are classified into lateral, middle, and medial tumors from an anatomical perspective [5]. Surgery of medial sphenoid wing meningiomas (MSWMs) can be challenging, particularly in terms of preserving essential neurological (e.g., optic nerve, oculomotor nerve) and vascular structures (e.g., cavernous sinus, internal carotid, and middle cerebral arteries [6,7,8,9,10]. 

Against this backdrop, surgery for MSWMs can lead to serious morbidities, such as decreased vision, ischemic lesions, or diplopia due to a palsy of the cranial nerves (CN III, IV, and VI) innervating the extraocular muscles [11,12,13]. Postoperative oculomotor nerve palsies are reported with rates of 9.5% or 10.3% [14,15], whereas vision deterioration has been observed in up to 22% of cases after surgery for MSWM [16]. The risk of surgical morbidity in cranial meningioma surgery is significantly influenced by tumor texture, including characteristics such as adherence or infiltration of neurovascular structures and brain edema [17,18,19,20]. Recently, baseline tumor shape was found to be significantly correlated with the WHO grade, adding another factor to consider in surgical planning [21].

However, the impact of tumor shape on postoperative neurological functioning in surgery for MSWM and its impact on tumor progression have not been investigated yet. The present study aims to examine the role of tumor shape in postoperative morbidity, neuropathological characteristics, and progression-free survival (PFS). 

## 2. Materials and Methods

### 2.1. Study Design and Patient Characteristics

Between January 2010 and November 2021, 74 patients with primary sporadic MSWM underwent surgical treatment at the authors’ institution and were analyzed retrospectively. Inclusion criteria were histopathologically confirmed primary meningioma, age greater than 18 years, and sporadic meningioma. MSWMs were defined as meningiomas originating from the anterior clinoid and the medial third of the lesser sphenoid wing from an anatomical point of view [22]. Patients with neurofibromatosis type 2-associated meningioma or meningioma patients who underwent prior radiotherapy were excluded from further analysis due to their different neuropathology and proliferative potential [23]. The decision-making process was carried out by an interdisciplinary neuro-oncological board consisting of senior experts in the fields of neuro-oncology, neuroradiology, radiotherapy, and neurosurgery. Regular follow-up after surgery comprises first imaging at 3 months after surgery and further images on an annual basis for WHO grade 1 meningiomas and every 6 months for WHO grade 2 meningiomas [24].

### 2.2. Data Recording

Clinical characteristics, including demographics, cranial nerve functioning, neurological deficits, Karnofsky Performance Status (KPS), WHO grading based on neuropathological investigations, extent of meningioma resection based on the Simpson classification system in line with the European Association of Neuro-Oncology (EANO) (Simpson grades I–III constitute gross total resection, Simpson grade IV constitutes subtotal resection, and Simpson grade V constitutes biopsy), and postoperative follow-up data were recorded and entered into a computerized dataset (SPSS, Version 29 for Windows, IBM Corp., Armonk, NY, USA) [24]. Initial T1-weighted (native and gadolinium-enhanced) and T2-weighted high-field magnetic resonance (MR) imaging (3.0 Tesla) was routinely performed within 48 h before surgery for MSWM. Tumor shape was considered irregular if the edges were irregular, mushroom-shaped, lobulated, and the boundary with adjacent cortex was unclear [25]. The shape was judged independently by two reviewers (JW and FA). The shape of MSWM was further quantified using sphericity, which measures how closely an object’s shape resembles a perfect sphere. Sphericity was calculated as the ratio of the surface area of a sphere with the same volume as the object to the surface area of the object. To determine the sphericity of a meningioma, for instance, the sphericity of a meningioma can be determined by comparing the surface area of a sphere with the same volume as the tumor to the surface area of the meningioma itself. The Hakon Wadell method was used to calculate the sphericity, with the following formula [26]: Sphericity = π^1/3^ (6 × tumor volume)^2/3^/surface area. A value of 1 indicated a perfect sphere, whereas lower values indicated a more complex shape. Gadolinium-enhanced T1-weighted MR images were used to determine MSWM volume and MSWM surface area, which were calculated using volumetric analysis in 3D slicer software (version 5.2.1; https://www.slicer.org (accessed on 1 October 2022)) [27]. Segmentation was performed in a semi-automatic workflow with the “Fast Marching” method [28]. Figure 1 shows a representative case of an irregularly shaped MSWM with very low sphericity and another case with a more regular shape and a higher sphericity. A radiomic imaging examination was performed retrospectively by a blinded observer regarding the pathological and clinical data of the patients. Peritumoral edema was defined as hyperintensity adjacent to the meningioma on T2-weighted MR images [29].

Tumor consistency was graded as soft or firm using Zada’s consistency grading system [30]. Soft MSWMs were defined as those amenable to being resected totally or mainly with suction (constituting Grades 1 and 2), whereas firm MSWMs were defined as those requiring sharp resection, ultrasonic aspiration, or calcified portions (constituting Grades 3, 4, or 5). Pial vascularization of the MSWM was evaluated using T2-weighted MR images, and peritumoral flow voids were graded as present or absent [31]. The integrity of the arachnoid layer was investigated using T2-weighted MR images and was classified as intact in the presence of cerebrospinal fluid at the brain/MSWM surface [25].

### 2.3. Surgical Procedure and Follow-Up Regime

Surgical resection was performed under general anesthesia using white-light resection guided by neuronavigation (Brainlab Curve, BrainLAB AG, Feldkirchen, Germany) and an operating microscope (Pentero, Carl Zeiss, Jena, Germany). Microscopic resection was performed using either the frontolateral or the pterional approach. In cases where MSWMs had infiltrated the cavernous sinus, the intracavernous tumor portion was not completely resected to prevent new-onset cranial nerve dysfunctions. Clinical and imaging follow-up appointments included MR images taken 3 months after MSWM surgery and annually for the subsequent 5 years. Earlier examinations were scheduled in cases of new or worsening neurological deficits or neuroradiological signs of MSWM progression. Meningioma progression was defined as the appearance of new local MSWM lesions or the growth of progressive residual MSWMs at the site of the initial surgery on a follow-up MRI examination [32].

### 2.4. Immunohistochemistry

Paraffin sections were stained with hematoxylin and eosin (H&E). Immunohistochemical analysis was performed on the sections using the Molecular Immunology Borstel-I (MIB-I, DAKO, Glostrup, Denmark) antibody to detect the Ki67 antigen and determine the MIB-1 index by the average method. 

### 2.5. Statistics of Institutional Data

Fisher’s exact test (two-sided) was used to compare nominal variables and the Student’s *t*-test for metric variables between patients with regularly shaped or irregularly shaped MSWMs, patients with normal or increased MIB-1 labeling index, and patients with postoperative cranial nerve integrity or new cranial nerve deficits. Radar charts were created to visualize the univariate comparisons. We only reported two-sided *p*-values. We performed log-rank tests, and Kaplan–Meier charts of PFS were created using GraphPad Prism 8 (GraphPad Software, San Diego, CA, USA). A *p*-value < 0.05 was defined as statistically significant. Cut-off values of MIB-1 index and tumor volume in predicting tumor progression were identified using receiver operating characteristic (ROC) curve analysis. A multivariable Cox regression analysis was conducted to evaluate the PFS, and multivariable binary logistic regression analyses were conducted to investigate factors influencing new cranial nerve deficits and an increased MIB-1 labeling index. The data were organized and analyzed using SPSS© (Version 29, IBM Corp., Armonk, NY, USA) and R software 4.2.2 (R Foundation for Statistical Computing, Vienna, Austria).

### 2.6. Systematic Review

#### 2.6.1. Search Workflow

We conducted a systematic literature review using the MEDLINE database (accessed in January 2023). We searched for the following keywords, either individually or in relevant combinations: “meningioma”, “shape”, “Ki-67”, and “MIB-1”. Full-text articles were retrieved from all results and independently reviewed for potential inclusion by two authors (JW and MV). Any disagreements between these authors were resolved by consensus with two additional authors (FA and EG). References to the relevant articles were also screened for additional studies of interest.

#### 2.6.2. Selection Criteria

We analyzed articles about patients with regularly or irregularly shaped cranial meningiomas as well as their references. We included articles that analyzed and provided detailed individual data about the shape and mean ± standard deviation values of the MIB-1/Ki-67 index. We did not screen for progression-free survival data due to significant changes in the WHO classification system over time, which would have made reliable statistical analysis impossible [4]. We excluded case series and case reports with limited data on a small number of subjects to avoid super-selection bias.

#### 2.6.3. Data Collection, Data Extraction, and Statistical Analysis

We extracted data on patient characteristics, study design, definition of tumor shape, proportions of regularly and irregularly shaped tumors, and MIB-1 labeling indices, including mean values and corresponding standard deviations. The data were entered into SPSS© (Version 29, IBM Corp., Armonk, NY, USA). To investigate statistical heterogeneity and inconsistency, we used x^2^ and I^2^ statistics, respectively. An I^2^ value of 50% or more represented substantial heterogeneity [33]. We also considered the weight of the relative contribution of individual studies based on sample size. To assess publication bias, we used two methods: (1) creating funnel plots to visually examine the publication bias of included studies; and (2) using an Egger regression test to statistically investigate the symmetry of the funnel plot. The two-tailed Egger regression intercept test was set to a 5% significance threshold to determine the likelihood of publication bias [34]. To account for heterogeneity, we used a random effect model. The overall mean difference in MIB-1 labeling index between regularly and irregularly shaped cranial meningiomas was calculated.

## 3. Results

### 3.1. Patient Characteristics

We analyzed data from 74 patients who met the inclusion criteria. The median age (±SD) was 59.0 years (±14.1 years), and there was a female predominance (female:male ratio = 1.85:1.0) in the patient population. The median KPS (interquartile range (IQR)) was 80.0 (70.0–80.0). Preoperative cranial nerve dysfunctions were recorded, and preoperative visual deterioration was observed in 26 patients (35.1%). Preoperative dysfunctions of the cranial nerves III, IV, and VI were observed in seven (9.5%), ten (13.5%), and eight (10.8%) patients, respectively. Cavernous sinus infiltration was present in 16 (21.6%) patients, and peritumoral brain edema was present in 27 (36.5%) cases, with a mean (±SD) edema volume of 15.5 ± 12.6 cm^3^. The mean volume (±SD) and sphericity (±SD) of MSWM were 23.4 ± 28.8 cm^3^ and 0.75 ± 0.17, respectively. WHO grades 1 and 2 MSWMs were observed in sixty-eight (91.9%) and six (8.1%) patients, respectively. The mean (±SD) MIB-1 labeling index was 3.9 ± 3.1. Adjuvant radiation therapy was performed in 6 patients (8.5%). Further details are provided in Table 1.

### 3.2. Patient Characteristics in Regularly and Irregularly Shaped Medial Sphenoid Wing Meningiomas

Regular MSWM shape was present in 43 cases (58.1%), and irregular shape was observed in 31 cases (41.9%), respectively. The median age (±SD) at diagnosis was 60.5 ± 11.4 years for patients with regularly shaped MSWM, and 58.6 ± 17.3 years for those with irregular MSWM (*p* = 0.60). KPS and sex were homogeneously distributed between the groups. Preoperative cranial nerve (CN) III palsy was observed in eight patients (25.8%) with irregularly shaped MSWMs, and in two patients (4.7%) with regularly shaped MSWM, respectively (*p* = 0.014). There was no significant difference in preoperative rates of CN II and CN IV deterioration between the two groups. Irregularly shaped MSWM infiltrated the cavernous sinus in nine cases (29.0%), whereas cavernous sinus infiltration was present in seven cases (16.3%) with a regularly shaped MSWM (*p* = 0.25). The presence of calcification and vascular encasement did not differ significantly between the two groups (Figure 2A). 

The mean (±SD) surface area in regularly shaped MSWM was 2678.5 ± 2144.9 mm^2^, while the mean (±SD) surface area was 5334.1 ± 4277.1 mm^2^ in irregularly shaped MSWMs (*p* = 0.004). Tumor volume (mean (±SD) tumor volume: 18.9 ± 22.9 cm^3^ (regular shape) vs. 29.7 ± 34.9 cm^3^ (irregular shape); *p* = 0.14) and brain edema volume (mean (±SD) brain edema volume: 13.9 ± 10.9 cm^3^ (regular shape) vs. 19.2 ± 16.3 cm^3^ (irregular shape); *p* = 0.33) did not differ significantly between regularly and irregularly shaped MSWMs. Moreover, patients with regularly shaped MSWM underwent Simpson grade ≤II resections more frequently (34/43; 79.1%) compared with those with irregularly shaped MSWMs (16/31; 51.6%; *p* = 0.02). Sphericity, which is a variable used to quantify shape (Figure 2B), was significantly lower in MSWMs with an irregular shape (0.81 ± 0.14 vs. 0.69 ± 0.18, *p* = 0.006). The mean MIB-1 labeling index was 5.0 ± 3.8 in patients with irregularly shaped MSWM, and 3.2 ± 2.2 in those with regularly shaped MSWMs, respectively (*p* = 0.009. characteristics for regularly and irregularly shaped MSWM are summarized in Table 2. 

### 3.3. Progression-Free Survival Outcomes in Regularly and Irregularly Shaped Medial Sphenoid Wing Meningiomas

The mean follow-up time for the entire cohort was 54.0 months ± 47.5 months. MRI surveillance follow-up data were available for 67 out of 74 patients (90.5%). The median time to meningioma progression in the entire cohort was 138.0 months (95% CI: 114.5–161.5). Patients with an irregularly shaped MSWM had a mean PFS time of 97.0 months (95% CI = 76.3–117.8, *n* = 30), whereas those with a regularly shaped MSWM had a mean PFS time of 132.1 months (95% CI = 121.5–142.7, *n* = 37). The Kaplan–Meier method showed that irregular shape was significantly associated with a shortened PFS time compared with regularly shaped MSWM patients (log-rank test: *p* = 0.007, Figure 3A). 

Univariate Cox regression analysis revealed a significant association between irregular MSWM shape and tumor progression (HR: 6.23, 95% CI = 1.37–28.33, *p* = 0.018). Optimal cut-off values for MIB-1 labeling index and tumor volume were identified using a ROC curve analysis. The area under the curve (AUC) for tumor volume and MIB-1 index in predicting tumor progression were 0.76 (95% CI: 0.63–0.88, sensitivity: 76.9%, specificity: 70.9%) and 0.88 (95% CI: 0.79–0.96, sensitivity: 84.6%, specificity: 80.0%), respectively. Optimum cut-off values for tumor progression estimation were identified as preoperative tumor volume (≥9.875/<9.875 cm^3^) and MIB-1 index (≥5%/<5%). Univariate analysis revealed an additional significant association between increased MIB-1 index (≥5%) and shortened PFS time. The mean PFS of MSWM patients with an increased MIB-1 index (≥5%) was 84.1 months (95% CI = 58.7–109.5), whereas those with a low MIB-1 index (<5%) had a mean PFS time of 124.7 months (95% CI = 119.3–130.1) (Univariate Cox regression analysis: HR = 9.49, 95% CI = 2.06–43.77, *p* = 0.004). Univariate Cox regression analyses also found significant associations between the presence of brain edema and shortened time to MSWM progression (mean PFS times: 81.3 (95% CI: 60.8–101.8) vs. 129.2 (95% CI: 115.2–143.1); HR = 4.85, 95% CI = 1.39–16.95, *p* = 0.013).

A multivariate cox regression analysis was performed to investigate the association of PFS with shape (irregular/regular), brain edema (presence/absence), MIB-1 index (≥5%/<5%), Simpson grade (≥III/<III), tumor volume (≥9.875 cm^3^/<9.875 cm^3^), and cavernous sinus invasion (presence/absence). The analysis identified irregular shape as an independent statistically significant predictor for shortened PFS in MSWMs (HR = 8.03, 95% CI = 1.04–62.10, *p* = 0.046). Brain edema (HR = 12.76, 95% CI = 1.76–92.57, *p* = 0.012) and MIB-1 index ≥5% (HR = 7.25, 95% CI = 1.11–47.20, *p* = 0.038) were also found to be significantly associated with shortened PFS. The result of the multivariate Cox regression analysis is displayed in Figure 3B. 

### 3.4. Association between Shape and MIB-1 Index

The mean MIB-1 labeling index in the entire cohort was 3.9 ± 3.1%. An MIB-1 labeling index of ≥5% has been shown to predict MSWM progression. Therefore, the cohort was dichotomized using the identified cut-off value, as shown in the frequency distribution histogram in Figure 4A. Fifty patients (67.6%) had a normal MIB-1 index (<5%), and twenty-four patients had an increased (≥5%) MIB-1 index (32.4%). Patient characteristics were compared between those with a normal (<5%) or increased (≥5%) MIB-1 index. Fourteen patients with a firm MSWM had an increased MIB-1 index (14/24; 58.3%), and twenty-three patients with a firm MSWM had a normal MIB-1 index (23/50; 46.0%), respectively (*p* = 0.46). Fourteen patients with a peritumoral flow void suggesting pial blood supply had a normal MIB-1 index (14/50; 28.0%), whereas ten patients with a pial blood supply had an increased MIB-1 index (10/24; 41.7%; Fisher’s exact test (two-sided): *p* = 0.29). Fifteen MSWM patients (15/50; 30.0%) with a normal MIB-1 index had a disrupted arachnoid layer, and eleven MSWM patients (11/24; 45.8%) with an increased MIB-index had a disrupted arachnoid layer, respectively (*p* = 0.20). Eighteen MSWM patients (18/50; 36.0%) with a normal MIB-1 labeling index were male, and eight MSWM patients (8/24; 33.3%) with an increased MIB-1 labeling index were male, respectively (*p* = 0.99). Patients with an increased MIB-1 index had a mean age of 63.2 ± 16.0 years old, whereas patients with a normal MIB-1 index had a mean age of 58.0 ± 13.0 years old (*p* = 0.14). Increased MSWM volume (≥9.875 cm^3^) was observed in 15 patients (15/24; 62.5%) with an increased MIB-1 index, and in 24 patients (24/50; 48.0%) with a normal MIB-1 index, respectively (*p* = 0.32). Of those with an infiltration of the cavernous sinus, four cases (4/24; 16.7%) had an increased MIB-1 index, while twelve MSWM patients (12/50; 24.0%) with a cavernous sinus infiltration had a normal MIB-1 index (*p* = 0.56). An elevated MIB-1 index was observed in 17 cases (17/24; 70.8%) of those patients who underwent a Simpson grade I or II resection, whereas 33 MSWM patients (33/50; 66.0%) of those with a Simpson grade I or II resection had a normal MIB-1 index (*p* = 0.79). Brain edema was observed in 54.2% (13/24) of MSWM patients with an increased MIB-1 index and in fourteen MSWM patients (14/50; 28.0%) of those with a normal MIB-1 index, respectively (*p* = 0.04). MSWMs with an irregular shape were found in 62.5% (15/24) of cases with an increased MIB-1 index and in 32.0% (16/50) of MSWMs with a normal MIB-1 index, respectively (*p* = 0.02). Figure 4B shows a radar plot summarizing the patient- and disease-related characteristics of surgically treated MSWM patients with a normal (<5%) or increased (≥5%) MIB-1 labeling index. 

The mean (±SD) MIB-1 labeling index in those with a regularly shaped MSWM was 3.16 ± 2.25, whereas the mean MIB-1 labeling index in those with an irregularly shaped MSWM was 5.03 ± 3.75 (*p* = 0.009). Figure 4C displays violin plots that summarize and compare the distribution of MIB-1 labeling indices between patients with regularly and irregularly shaped MSWMs. Multivariate binary logistic regression analysis included the following variables: sex (male/female), brain edema (present/absent), MSWM shape (irregular/regular), tumor volume (≥/<9.875 cm^3^), and Simpson grade (≤II/>II). The multivariate analysis revealed that brain edema (OR: 5.28, 95% CI: 1.31–21.38, *p* = 0.02) and irregular shape (OR: 7.59, 95% CI: 2.04–28.25, *p* = 0.003) are independently associated with an MIB-1 labeling index ≥5% in MSWMs. Figure 4D displays forest plots summarizing the results of the multivariate binary logistic regression analysis.

### 3.5. Association between Shape and New Cranial Nerve Morbidity after Medial Sphenoid Wing Meningioma Surgery

Eleven (14.9%) patients developed new cranial nerve deficits after surgery for MSWM. Among the eleven patients with new cranial nerve deficits, eight (8/11; 72.7%) had a new palsy of the third cranial nerve. In two cases (2.7%), involvement of the optic nerve was observed, and in three cases (4.1%), involvement of the trochlear nerve was observed. Two cases presented simultaneous involvement of the third cranial nerve combined with dysfunctions of the optic nerve or the trochlear nerve. Patients with a new postoperative cranial nerve deficit were more likely to have an irregular-shaped MSWM (8/11; 72.7%) than patients with postoperative cranial nerve integrity (23/63; 36.5%; *p* = 0.04). Increased tumor volume (≥9.875 cm^3^) was found in three patients (3/11; 27.3%) with a new cranial nerve deficit and in 36 (36/63; 57.1%) patients without a new cranial nerve deficit, respectively (*p* = 0.10). Firm MSWM consistency was present in five patients (5/11; 45.5%) with new cranial nerve deficits, whereas thirty-two (32/63; 50.8%) patients had a firm MSWM and had no new cranial nerve deficits postoperatively (Fisher’s exact test (two-sided): *p* = 0.99). Two patients (2/11; 18.2%) with a pial blood supply had new postoperative cranial nerve deficits, and twenty-two patients (22/63; 34.9%) with a pial blood supply had no new cranial nerve deficits, respectively (*p* = 0.49). Disruption of the arachnoid layer was present in six (6/11; 54.5%) patients with new cranial nerve deficits, and disruption of the arachnoid layer was observed in twenty (20/63; 31.7%) patients without a new postoperative cranial nerve deficit, respectively (*p* = 0.18). Preoperative brain edema was present in three patients (3/11; 27.3%) with new cranial nerve deficits, whereas twenty-four (24/63; 38.1%) MSWM patients had preoperative brain edema with postoperative cranial nerve integrity (*p* = 0.74). Calcified MSWM (1/11; 9.1%) was observed in one patient with a postoperative new cranial nerve deficit and in five patients (5/63; 7.9%) of those without a new cranial nerve deficit (*p* = 0.99). Infiltration of the cavernous sinus was found in four (4/11; 36.4%) patients with a new postoperative cranial nerve deficit compared with twelve patients (12/63; 75.0%) who had MSWM with cavernous sinus infiltration and postoperative cranial nerve integrity (*p* = 0.24). Simpson grade I or II resections were performed in six (5/11; 54.5%) patients with a new postoperative cranial nerve deficit, whereas Simpson grade I or II resections were performed in forty-four (44/63; 69.6%) patients with postoperative cranial nerve integrity (*p* = 0.32). Arterial vascular encasement was present in five MSWMs with postoperative new cranial nerve deficits (5/11; 45.5%) and in twenty-eight MSWMs without a new postoperative cranial nerve deficit (28/63; 44.4%), respectively (*p* = 0.99). Figure 5A displays a radar chart summarizing the results of the univariate analysis. 

A multivariate binary logistic regression analysis with consideration of the following variables was performed: Simpson grade (≤II/>II), cavernous sinus infiltration (present/absent), calcification (present/absent), brain edema (present/absent), vascular arterial encasement (present/absent), tumor volume (≥/<9.87 cm^3^), and shape (irregular/regular). Multivariate logistic regression analysis revealed that irregular shape was significantly associated with new postoperative cranial nerve deficits (OR = 5.75, 95% CI = 1.15–28.63, *p* = 0.03). Figure 5B shows forest plots summarizing the results of the multivariate binary logistic regression analysis.

### 3.6. Association between Shape and Postoperative Functioning of Preoperatively Exisiting Cranial Nerve Deficits

Twenty-six patients had either a single or a combined preoperative cranial nerve deficit of the cranial nerves II, III, IV, or VI. Postoperatively, five of those patients (5/26; 19.2%) had a postoperative worsening of the preoperatively existing cranial nerve deficit. There were three further deteriorations of the visual acuity (CN II) and two worsenings of preoperatively existing oculomotor nerve palsies. One of the postoperatively further deteriorating cranial nerve palsies was observed in a regularly shaped MSWM (1/14; 7.1%), whereas four worsenings were observed in irregularly shaped MSWMs (4/12; 33.3%; Fisher’s exact test (two-sided): *p* = 0.15). 

### 3.7. Systematic Review

#### 3.7.1. Literature Search Results of Meningioma Shape and MIB-1 Index

The PubMed search identified a total of 488 titles, of which 18 were deemed relevant after removing duplications and applying the mentioned inclusion criteria (see Appendix A). Two manuscripts reporting on a total of 210 patients met the inclusion criteria after further screening of the remaining articles [35,36]. Both included studies were retrospective and classified as level III evidence. In total, the pooled analysis, including the current series, comprised 284 patients to investigate the association between shape and the MIB-1 index in cranial meningiomas. Patient characteristics of the included studies are summarized in Table 3. 

#### 3.7.2. Association between Shape and MIB-1 Index in the Pooled Analysis

A total of 210 patients were included in this analysis. Patients with an irregularly shaped meningioma had a higher MIB-1 index, with a mean difference of 1.98 (95% CI: 1.38–2.59), compared with patients with regularly shaped meningiomas (*p* < 0.01, see Figure 6A). Publication bias was assessed using a funnel plot and Egger’s regression-based test (see Figure 6B). All included study data points were located inside the inverted funnel, indicating a small publication bias regarding the analysis of the association between shape and the MIB-1 index in cranial meningiomas. The Egger’s regression-based test showed no statistically significant publication bias (*p* = 0.48, intercept = 1.129, 95% CI = −12.259–14.518).

## 4. Discussion

In our study cohort of 74 sporadic primary medial sphenoid wing meningiomas, we investigated the impact of tumor shape on postoperative cranial nerve functioning, proliferative potential, and progression-free survival. Our findings can be summarized in four key points. First, the shape of MSWMs can be objectively quantified using the shape features “sphericity” or “surface area”. Second, the irregular shape of MSWMs is significantly associated with a shorter time to tumor progression. Third, the irregular shape of MSWMS may result from increased proliferative activity, as measured by a higher MIB-1 labeling index. This assumption is supported by the current series and the results of the pooled data analysis. Fourth, patients with irregularly shaped MSWM are at a higher risk of developing new postoperative cranial nerve deficits.

The shape of meningiomas is often classified as regular or irregular based on their margins and infiltrative patterns. Irregular shape is thought to result from substantial heterogeneity in proliferative activity in various subregions of the tumor, with some areas exhibiting significantly increased growth rates [37,38]. Thus, meningioma shape has been suggested as a means to estimate the meningioma grade preoperatively [39]. Sphericity is a quantitative shape feature that can be extracted from clinical images and has been found to be correlated with endpoints such as overall survival in glioblastoma [40]. In a retrospective review of 303 patients who underwent resection of WHO grades 1–3 non-skull or skull base meningiomas, low sphericity was found to be a significant predictor of local tumor recurrence and overall survival [41]. The present investigation is the first to exclusively evaluate shape in the anatomical subgroup of medial sphenoid wing meningiomas, and we have identified that the independent subjective review of MSWM shape can be objectively quantified by the imaging-derived factors sphericity and surface area. 

We found that irregularly shaped MSWMs are an independent risk factor for a shortened time to tumor progression. There may be several reasons to explain this finding: (1) Irregular shape may result from increased proliferative potential, which can enhance tumor progression [37,38]. This finding is consistent with our study’s results, which identified irregular shape as an independent preoperative imaging-derived risk factor, estimating an increased MIB-1 labeling index in MSWMs. (2) Irregular shape has been shown to correlate with WHO grade 2 in other studies. However, our study exclusively analyzes MSWMs and includes only six WHO grade 2 meningiomas, which is too small to adequately address this issue [21,35,37,39,42]. (3) Irregularly shaped MSWMs are significantly more challenging to operate and are at an increased risk of subtotal resection compared with regularly shaped MSWMS. This finding has been described by Musigmann et al. [43], who performed a retrospective review using radiomics and machine learning to assess the preoperative risk of subtotal resection in skull meningiomas. Skull-base meningiomas or posterior fossa meningiomas with an irregular shape are at an increased risk of incomplete resection. In the present study, we also found that irregularly shaped MSWMs were significantly less likely to undergo a Simpson grade ≤II resection compared with those with a regular shape. Simpson grade ≤II resection was performed in 67.6% of the patients in the present cohort. Cavernous sinus infiltration was the major limitation regarding the achievement of a gross total resection in a higher proportion of patients. Hence, the rates of gross total resection of medial sphenoid wing meningiomas also range between 5% and 80% in the literature [5,44,45,46]. However, all those series had different proportions of patients with or without cavernous sinus infiltration, making them not entirely comparable. Postoperative radiotherapy has been shown to significantly improve PFS time in cases with cavernous sinus infiltration and residual tumor [14]. However, in the present series, only six patients underwent adjuvant radiation therapy for MSWM, which is a too small number to draw conclusions regarding the role of radiation therapy in tumor progression.

The MIB-1 labeling index (</≥5%) has been identified as a factor predicting the probability of PFS in medial sphenoid wing meningiomas. It is a known factor that influences PFS in cranial meningiomas [47,48]. The MIB-1 labeling index in sporadic cranial meningiomas is influenced by several factors, such as the anatomical location of the meningioma, sex, the density of CD68^+^ macrophage infiltrates, cyclooxygenase-2 (COX-2) expression, autocrine inhibitory inflammatory regulation, and progesterone receptor status [47,49,50,51,52]. In the present investigation, we observed no association between sex and an increased MIB-1 index. Previous investigations reported a positive correlation between male sex and higher MIB-1 labeling indices [53,54]. Matsuno et al. [54] performed an immunohistochemical analysis to investigate the proliferative potential of 127 meningioma patients. They found that male patients had an average MIB-1 index of 5.5%, while female patients had an average index of 2.7%. Furthermore, in the present investigation, we have no data regarding the inflammatory microenvironment or the progesterone receptor status. Future studies should investigate whether these known factors influencing the MIB-1 labeling index also affect imaging-derived factors that describe shape (e.g., sphericity, surface area). The present literature review confirmed that our findings are consistent with other studies showing a correlation between irregular shape and the MIB-1 labeling index, indicating that shape may serve as a potential quantitative imaging-derived feature to estimate the MIB-1 labeling index. In addition, anti-inflammatory drugs that selectively or non-selectively inhibit COX-2 have been demonstrated to reduce tumor growth and the MIB-1 labeling index [55,56]. However, the molecular classification of meningiomas according to their shape remains unknown. Hence, future studies should investigate whether meningioma shape can be assigned to one of the integrative molecular meningioma groups classified by Nassiri et al. [57]. Homozygous focal deletions of the cyclin-dependent kinase inhibitor 2A (CDKN2A) gene have been observed in highly proliferative meningiomas, such as anaplastic WHO grade 3 meningiomas [58,59,60]. CDKN2A/B homozygous deletions on chromosome 9p21 have been demonstrated to be predictive regarding progression-free survival in WHO grade 2 and 3 meningiomas [60]. However, in the series of Sievers et al. [60], analyzing 528 patients, only 4.9% had a homozygous CDKN2A/B deletion. Hence, other characteristics, such as mutations of the TERT promoter, will have to be investigated in irregularly shaped meningiomas to further understand the molecular pathophysiology of meningioma shape. A meta-analysis investigating data from 677 patients revealed that TERT alterations predicted progression-free survival independent of the WHO grades in meningiomas [61]. Furthermore, loss of the NF2 suppressor gene at chromosome 22 has been demonstrated in a higher percentage (50–60%) of meningiomas, and supratentorial meningiomas with NF2 alterations had higher MIB-1 indices and a shorter time to tumor progression [62,63]. Those emerging molecular markers and their association with shape will be of importance in future studies.

Postoperative new cranial nerve deficits were significantly associated with the presence of an irregular MSWM shape. In the present investigation, new postoperative cranial nerve deficits were observed in 14.9% of the patients, with the oculomotor nerve being the most frequently affected cranial nerve in the postoperative period. Despite significant advancements in skull base surgery over the last few decades, managing meningiomas in the middle cranial fossa still presents a major challenge due to their proximity to brain nerves and critical blood vessels, which can result in postoperative cranial nerve deficits that significantly impact the patient’s quality of life [64,65,66]. Therefore, maintaining cranial nerve functioning is a crucial quality measure in skull base meningioma surgery. The arachnoid membrane partitions meningiomas from normal brain parenchyma. Hence, this arachnoid layer is sometimes disrupted and can make meningioma surgery challenging in critical locations with adjacent neurovascular structures [67]. Al-Mefty already described the importance of the arachnoid layer in his main principles of skull base surgery: Remove bone to minimize injury to the brain; respect the arachnoid plane; repair vessels with meticulous care; plan for reconstruction and closure with the opening; be vigilant to displace neurovascular anatomy; and strive for total removal during the first surgery as far as possible [22]. In the multivariate analysis, it was found that irregularly shaped MSWMs are an independent and significant predictor of postoperative cranial nerve deficits. This may be due to their increased MIB-1 labeling index and inflammatory macrophage invasion profile, leading to stronger immunological responses during surgery [68]. Stronger inflammatory responses might be observed in those MSWM cases with both an irregular shape and an increased MIB-1 index compared with those MSWMs that exhibit a significantly lower proportion of cellular components of the monocytic cell lineage. Immune cell infiltration is a key feature of meningiomas, with up to 30% of the cells being immune cells, which makes the meningioma microenvironment a complex landscape of interactions between tumor and immune cells [69,70]. Previous studies have demonstrated an association between shape and PFS in meningiomas, but this is the first report linking increased MIB-1 labeling index to postoperative cranial nerve functioning [41]. In terms of pathophysiology, local residual cells with increased proliferative activity driven by an inflammatory response can disturb the local functioning of the cranial nerves by contributing to the formation of scars in irregularly shaped MSWMs. Scar formation is mediated by inflammation, and the irregular boundaries of these tumors, as well as potential adhesions to cranial nerves, can make surgical dissection more challenging with regard to the differentiation from the healthy tissue. COX-2 plays a key role in scar formation, a well-known consequence of prolonged inflammatory responses [71]. Simultaneously, COX-2 expression has been suggested as a relevant parameter for evaluating meningioma proliferation, and positive correlations have been observed between higher MIB-1 indices and the presence of CD68-positive macrophages [47,50]. 

Further research is needed to explore the potential relationship between proliferative activity and COX2 expression as well as the potential therapeutic implications of inhibiting this pathway postoperatively in those with an irregularly shaped MSWM. Preclinical studies have shown that acetylsalicylic acid can have a beneficial effect in reducing demyelination and increasing the diameter of myelin sheaths during the regeneration of axons after peripheral nerve injury associated with inflammatory responses [72]. Furthermore, shape might also be a further variable in the follow-up care and the decision-making process to re-treat or perform a watch-and-wait policy. Machine learning algorithms using quantitative characteristics from medical images are an emerging field in neuro-oncology. To date, there are investigations reporting the benefit of preoperative MR radiomics using support vector machines to predict tumor progression in meningiomas [73]. Shape-related features were reported to be the most robust and reproducible. However, the current evidence in the field of meningiomas is predominantly based on the evaluation of the predictive value of preoperative radiomics regarding progression-free survival. Chemotherapy and radiotherapy can produce treatment-related effects that might mimic tumor progression and are a diagnostic challenge. A recent meta-analysis suggested that machine-learning imaging applications are useful in separating true tumor progression from treatment-related effects in gliomas and brain metastases [74]. Future studies will have to evaluate whether those algorithms are also transferrable to the postoperative follow-up images of WHO grade 2 or 3 meningioma patients, who also undergo adjuvant therapies.

The main limitation of this investigation is its retrospective and monocentric nature, with limited follow-up and a small number of patients. Determining the MIB-1 index in CNS tumors can be challenging due to sampling limitations, especially when meningiomas are partially or subtotally removed, which may not accurately reflect the area of highest proliferative activity [75]. Additionally, interobserver variability should also be considered when determining the MIB-1 index. Sphericity as a quantitative measure to evaluate the shape of meningiomas has to be further validated. Despite the significant difference between regularly and irregularly shaped MSWMs, we still observed a few cases with irregular borders and a comparatively high sphericity. Furthermore, our literature search did not allow for an analysis of the correlation between shape and progression-free survival, which would be more clinically relevant than the MIB-1 Index alone. Future studies should also consider the density of inflammatory cell infiltrates in relation to meningioma shape to validate our findings. 

## 5. Conclusions

The present study shows that an irregular MSWM shape is independently associated with an increased risk of new postoperative cranial nerve deficits and a shortened time to tumor progression. Further validation of this finding through multicentric studies is desirable. Irregular MSWM shapes may be caused by highly proliferative tumors with an increased MIB-1 index.

## Figures and Tables

**Figure 1 cancers-15-03096-f001:**
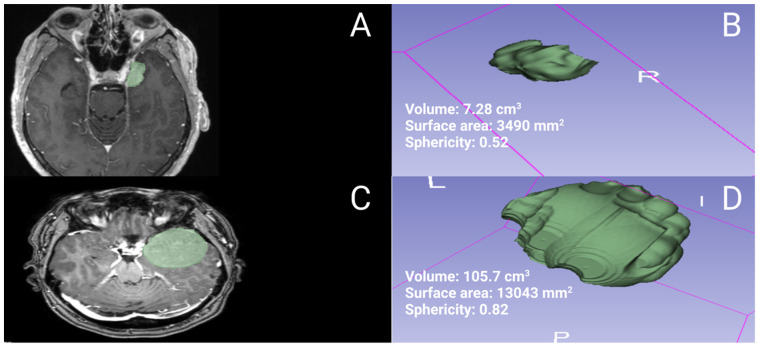
The figure displays two illustrative cases of MSWMs, one with an irregular shape shown in (**A**,**B**) along with its corresponding volumetric segmentation and 3D model, and the other with a regular shape shown in (**C**,**D**).

**Figure 2 cancers-15-03096-f002:**
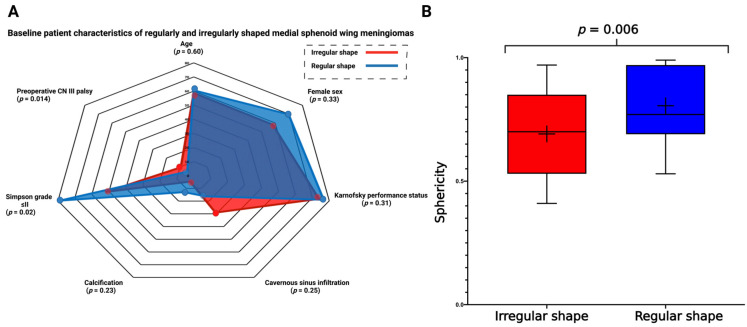
(**A**) Radar chart showing patient- and meningioma-related characteristics dependent on regular or irregular MSWM shape. (**B**) Box-plots displaying sphericity in patients with irregularly (red) or regularly (blue) shaped MSWMs. The whiskers display the minimum and maximum values. The vertical line in the box constitutes the median value and the plus symbol displays the mean value.

**Figure 3 cancers-15-03096-f003:**
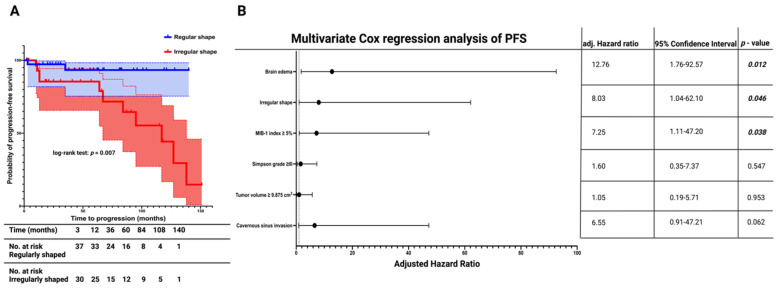
(**A**) Kaplan-Meier analysis (with the corresponding table showing number at risk table) of probability of progression-free survival stratified by regular (blue) and irregular (red) MSWM shapes. The corresponding shadowed fields represent the 95% confidence intervals of each Kaplan-Meier curve. Vertical dashes represent censored data (=progression-free at last visit) within both progression-free survival curves. Corresponding table displays the number of patients at risk in both groups at different time points. (**B**) Forest plots from multivariable Cox regression analysis: Irregular shape and MIB-1 index ≥5% are independent predictors of progression-free survival. *p*-values in bold and italics display statistically significant results.

**Figure 4 cancers-15-03096-f004:**
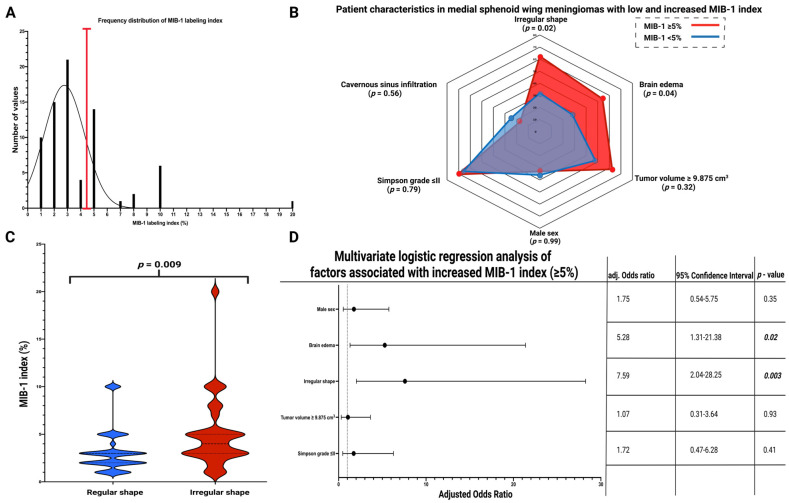
(**A**) Frequency distribution histogram for MIB-1 labeling index in the present cohort. The black bars indicate the number of MSWM patients with the corresponding MIB-1 labeling indices. The red vertical line shows the optimized cut-off point for MIB-1 labeling index regarding PFS. (**B**) Radar plot showing patient- and meningioma-related characteristics dependent on MIB-1 index ≥5% or <5%. (**C**) Violin plots displaying MIB-1 labeling indices in patients with irregularly (red) or regularly (blue) shaped MSWMs. (**D**) Forest plots from multivariable binary logistic regression analysis: Irregular shape and brain edema are independent predictors of increased MIB-1 labeling index in MSWMs. *p*-values in bold and italics display statistically significant results.

**Figure 5 cancers-15-03096-f005:**
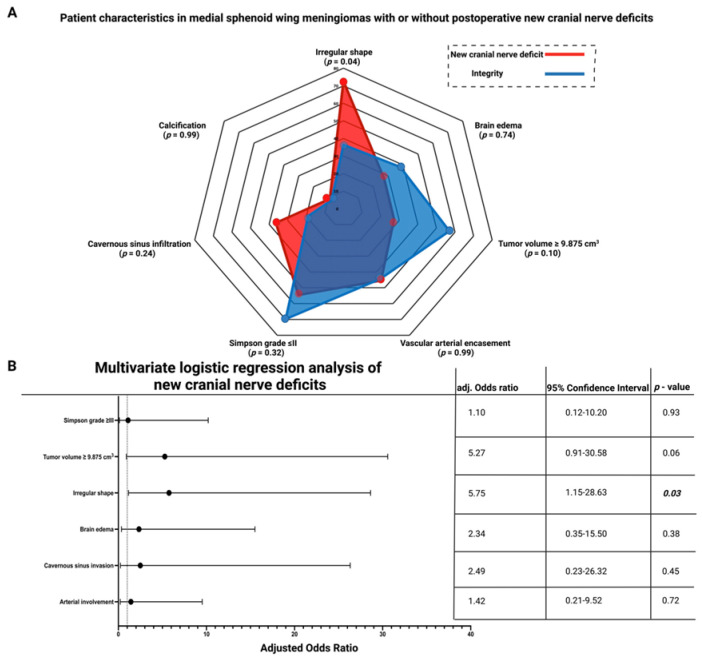
(**A**) Radar plot showing patient- and meningioma-related characteristics dependent on postoperative cranial nerve integrity or new postoperative cranial nerve deficits. (**B**) Forest plots from multivariable binary logistic regression analysis of predictors regarding new postoperative cranial nerve deficits: Irregular shape is an independent predictor of postoperative new cranial nerve deficits. *p*-values in bold and italics display statistically significant results.

**Figure 6 cancers-15-03096-f006:**
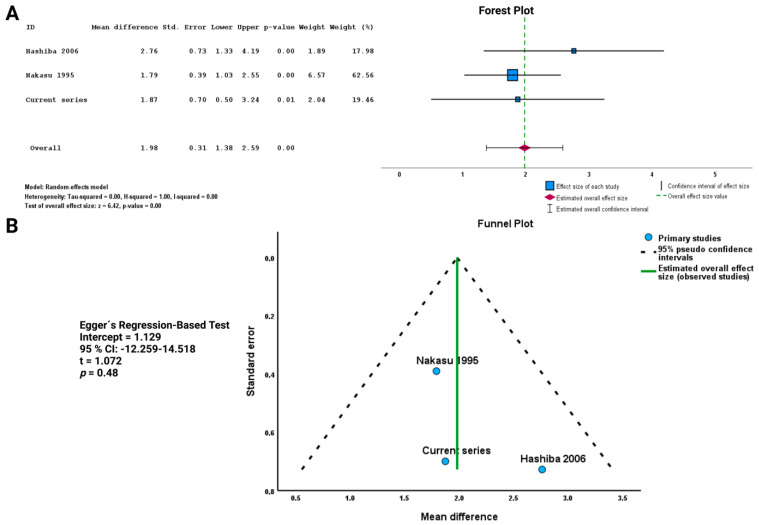
(**A**) Forest plots displaying the mean difference and the 95% CI estimates of the MIB-1 labeling index in studies comparing regularly and irregularly shaped meningiomas. (**B**) Funnel plots showing no publication bias. Results of Egger’s test analyzing publication bias are given [35,36].

**Table 1 cancers-15-03096-t001:** Patient characteristics.

Characteristics	*n* = 74
Median age (years, ±SD)	59.0 ± 14.1
Female sex	48 (64.9%)
Median preoperative KPS (IQR)	80.0 (70.0, 80.0)
Preoperative visual deterioration (CN II)	26 (35.1%)
Preoperative CN III dysfunction	7 (9.5%)
Preoperative CN IV dysfunction	10 (13.5%)
Preoperative CN VI dysfunction	8 (10.8%)
ASA classification	
I	8 (10.8%)
II	52 (70.3%)
III	13 (17.6%)
IV	1 (1.4%)
Cavernous sinus infiltration	16 (21.6%)
Vascular encasement	33 (44.6%)
Calcification	6 (8.1%)
Cystic appearance	6 (8.1%)
Peritumoral flow voids	24 (32.0%)
Disruption of arachnoid layer	26 (35.1%)
Brain edema	27 (36.5%)
Surface area, (mean ± SD), mm^2^	3881.8 ± 3519.7
Tumor volume, (mean ± SD), cm^3^	23.4 ± 28.8
Brain edema volume, (mean ± SD), cm^3^	15.5 ± 12.6
Sphericity, (mean ± SD)	0.75 ± 0.17
Simpson grade	
I	27 (36.5%)
II	23 (31.1%)
III	7 (9.5%)
IV	17 (23.0%)
Tumor consistency	
Soft	37 (50.0%)
Firm	37 (50.0%)
MIB-1 index, (mean ± SD),	3.9 ± 3.1
WHO grade	
1	68 (91.9%)
2	6 (8.1%)

Abbreviations: ASA = American Society of Anesthesiologists; CN = cranial nerve; IQR = interquartile range; KPS = Karnofsky Performance Status; MIB = Molecular Immunology Borstel; SD = standard deviation; WHO = World Health Organization.

**Table 2 cancers-15-03096-t002:** Comparison of patient characteristics between regularly and irregularly shaped medial sphenoid wing meningiomas (using Fisher’s exact test (two-sided) and independent *t*-test).

Characteristics	Regularly Shaped (*n* = 43)	Irregularly Shaped (*n* = 31)	*p*-Value
Median age (years ± SD)	60.5 ± 11.4	58.6 ± 17.3	0.60
Female sex	30 (69.8%)	18 (58.1%)	0.33
Median preoperative KPS (±SD)	76.7 ± 11.5	73.9 ± 12.3	0.31
Preoperative visual deterioration (CN II)	14 (32.6%)	12 (38.7%)	0.63
Preoperative CN III palsy	2 (4.7%)	7 (9.5%)	0.12
Preoperative CN IV dysfunction	2 (4.7%)	8 (25.8%)	*0.014*
Preoperative CN VI dysfunction	5 (11.6%)	3 (9.7%)	0.99
Cavernous sinus infiltration	7 (16.3%)	9 (29.0%)	0.25
Vascular encasement	17 (39.5%)	16 (51.6%)	0.35
Calcification	2 (4.7%)	4 (12.9%)	0.23
Cystic appearance	2 (4.7%)	4 (12.9%)	0.23
Pial blood supply	12 (27.9%)	12 38.7%)	0.45
Arachnoid layer disruption	11 (25.6%)	15 (48.4%)	0.052
Brain edema	19 (44.2%)	8 (25.8%)	0.14
Surface area, (mean ± SD), mm^2^	3112.9 ± 2736.6	5355.6 ± 4262.8	*0.02*
Tumor volume, (mean ± SD), cm^3^	18.9 ± 22.9	29.7 ± 34.9	0.14
Brain edema volume, (mean ± SD), cm^3^ (present in 27 cases)	13.9 ± 10.9	19.2 ± 16.3	0.33
Sphericity, (mean ± SD)	0.81 ± 0.14	0.69 ± 0.18	*0.006*
Simpson grade			0.10
I	18 (41.9%)	9 (29.0%)	
II	16 (37.2%)	7 (22.6%)	
III	2 (4.7%)	5 (16.1%)	
IV	7 (16.3%)	10 (32.3%)	
Tumor consistency			0.99
Soft	22 (51.2%)	15 (48.4%)	
Firm	21 (48.8%)	16 (51.6%)	
WHO grade			0.23
1	41 (95.3%)	27 (87.1%)	
2	2 (4.7%)	4 (12.9%)	
MIB-1 index, (mean ± SD)	3.2 ± 2.2	5.0 ± 3.8	*0.009*
Adjuvant radiation therapy	2 (4.7%)	4 (12.9%)	0.23
New cranial nerve deficit	3 (7.0%)	8 (25.8%)	*0.04*
Mean (95% CI) PFS time (months)	132.1 (121.5–142.7)	97.0 (76.3–117.8)	*0.007* ^a^

Italic numbers display statistically significant *p*-values. ^a^ = log-rank test; CN = cranial nerve; KPS = Karnofsky Performance Status; MIB = Molecular Immunology Borstel; PFS = progression-free survival; SD = standard deviation; WHO = World Health Organization.

**Table 3 cancers-15-03096-t003:** Summary of study characteristics.

Name/Year of Study	Study Design/Level of Evidence	Country	Definition/Measurement of Shape	No. Patients of Entire Cohort	No. Patients with Irregularly Shaped Meningioma	No. Patients with Regularly Shaped Meningiomas	Endpoints	MIB-1/Ki-67 Index	WHO Grade	Age	Sex
Hashiba et al. 2006 [35]	Retrospective/Level III	Japan	Shape of the tumor was classified as either smooth or irregular. The so-called mush-rooming tumors, i.e., tumors with fringes and a lobulated appearance, were considered irregular.	Total: 90	38	52	MIB-1	Regular: 1.82% ± 1.75% Irregular 4.58% ± 4.84%	79 WHO grade 1, 8 WHO grade 2 (atypical), 3 WHO grade 3 (anaplastic)	Mean age: 57.6 (range: 20–93)	Female:male ratio = 3.5:1
Nakasu et al. 1995 [36]	Retrospective study/Level III	Japan	Tumor shape was described as round or lobular. Round tumors had smoothly curved surfaces without notches, whereas lobular tumors showed globoid appearances with at least one notch.	120	26	94	MIB-1	Regular: 1.06 ± 0.67% Irregular: 2.85 ± 3.68%	107 WHO grade 1, 10 WHO grade 2 (atypical), 3 WHO grade 3 (anaplastic)	Mean age: 57.5 ± 13.2	Female:male ratio = 3:1
Present series	Level III	Germany	Tumor shape was considered irregular if the edges were irregular, mushroom-shaped, lobulated, and the boundary with adjacent cortex was unclear [25]. The shape was judged independently by two reviewers (JW & FA). The shape of MSWM was further quantified using sphericity, which measures how closely an object’s shape resembles a perfect sphere.	74	31	43	MIB-1, Cranial nerve deficits, PFS	Regular: 3.16 ± 2.25 Irregular: 5.03 ± 3.75	68 WHO grade 1, 6 WHO grade 2	Median age: 59.0 ± 14.1	Female:male ratio: 1.85:1

## Data Availability

All data are included in this manuscript.

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
