# Peer review of "Impact of Shape Irregularity in Medial Sphenoid Wing Meningiomas on Postoperative Cranial Nerve Functioning, Proliferation, and Progression-Free Survival"

_cancers, 2023, doi:10.3390/cancers15123096_

Round 1

Reviewer 1 Report

Good work.

Author Response

Dear Reviewer

Thank you for reading our manuscript and critically reviewing it, which will help us improve it to a better scientific level and make it more understandable to the readership.

Reviewer 2 Report

In this manuscript, the authors investigated the impact of tumor shape in medial sphenoid wing meningiomas (MSWM) by revisiting clinical data from 74 patients. They found that irregular shape was significantly associated with the new postoperative cranial nerve deficits and a shorter time to tumor progression. They also performed a systematic literature review, which similarly indicated the association of higher MIB-1 index with the irregular shape. Overall, the manuscript was well-written and may provide additional insights into the relationship between tumor shape and neuropathology, progression-free survival, and cranial nerve functioning. Some minor comments to consider are:

1. Are there any gender differences observed in this study or reported in the previous literature?

2. Although the authors suggested the irregular shape of MSWM is probably caused by increased proliferative activity, as evidenced by higher MIB-1 index level, they should include more comprehensive discussions on the molecular basis/mechanisms underlying what factors drive the formation of irregular tumor shape and why it leads to faster tumor progression, lower survival rate, etc.

3. How well can the tumor shape be used as a diagnostic/predictive marker for tumor progression or patient survival? Is it possible to develop related tools such as machine learning for this purpose? Are there any efforts from the literature on this aspect? It would be beneficial to include more discussions to help the readers better understand the potential clinical applications of this study.

Author Response

Dear Reviewer

Thank you for reading our manuscript and critically reviewing it, which will help us improve it to a better scientific level and make it more understandable to the readership.

In the following we would like to respond to your remarks:

  1. The reviewer is absolutely right that the important relationship between sex and tumor proliferation has to be further discussed. In the present subgroup of MSWMs we found no association between sex and proliferative potential in MSWMs. Furthermore, we could not identify an association between shape and sex. Unfortunately, the identified studies reporting meningioma shape and MIB-1 index do not report the proportions of male or female patients among the regularly or irregulary shaped meningiomas. Hence, we cannot analyze the data of the literature regarding the association between sex and shape. The simple association between male sex and increased MIB-1 indices has been reported in previous investigations [1, 2]. Matsuno et al. [1] immunohistochemically analyzed the proliferative potential of 127 meningioma patients and found that male patients had a mean MIB-1 index of 5.5%, whereas the female patients had a mean MIB-1 index of 2.7%. Further evidence for a sex difference is supported by the study of Kasuya et al. [2], which also showed that the male sex is an independent predictor of an increased MIB-1 index. Despite various literature reporting a link between male sex and elevated MIB-1 index, the finding is still highly debated because of the relationship between meningioma progression and female sex hormones [3, 4]. We added a paragraph debating this association to the section “4. Discussion”.
  2. We agree with the reviewer that future tailored care for meningioma patients necessitates the molecular investigation of the meningioma shape. Therefore, we revised the section “4. Discussion”. Homozygous focal deletions of the cyclin-dependent kinase inhibitor 2A (CDKN2A) gene have been observed in highly proliferative meningiomas such as anaplastic WHO grade 3 meningiomas [5, 6, 7]. CDKN2A/B homozygous deletions on the chromosome 9p21 has been demonstrated to be predictive regarding progression-free survival in WHO grade 2 and 3 meningiomas [8]. However, in the series of Sievers et al. [7] analyzing 528 patients only 4.9% had a homozygous CDKN2A/B deletion. Hence, also other characteristics such as mutations of the TERT promoter will have to be investigated in irregularly shaped meningiomas to further understand the molecular pathophysiology of meningioma shape. A meta-analysis investigating data from 677 patients revealed that TERT alterations predicted progression-free survival independent of the WHO grades in meningiomas [8]. Furthermore, loss of the NF2 suppressor gene at chromosome 22 has been demonstrated in a higher percentage (50-60%) of meningiomas and supratentorial meningiomas with NF2 alterations had higher MIB-1 indices and a shorter time to tumor progression [9, 10]. Those emerging molecular markers and their association with shape will be of importance in future studies. We strive to investigate the association between those molecular items with meningioma shape in a future study.
  3. Machine learning algorithms using quantitative characteristics from medical images are an emerging field in neuro-oncology. To date, there are investigations reporting the benefit of preoperative MR radiomics using support vector machine to predict tumor progression in meningiomas [11]. Shape-related features were reported to be the most robust and reproducible features. However, the current evidence in the field of meningiomas is predominantly based on the evaluation of the predictive value of preoperative radiomics regarding progression-free survival. Chemotherapy and radiotherapy can produce treatment-related effects, which might mimic tumour progression and are a diagnostic challenge. A recent meta-analysis suggested that machine-learning imaging applications are useful in the differentiation of true tumor progression from treatment-related effects in gliomas and brain metastasis [12]. Future studies will have to evaluate whether those algorithms are also transferrable to the postoperative care of WHO grade 2 or 3 meningioma patients, who also undergo adjuvant therapies and might suffer from early tumor progression necessitating an interdisciplinary decision-making progress regarding retreatment.

References

  1. Matsuno, A.; Fujimaki, T.; Sasaki, T.; Nagashima, T.; Ide, T.; Asai, A.; Matsuura, R.; Utsunomiya, H.; Kirino, T. Clinical and histopathological analysis of proliferative potentials of recurrent and non-recurrent meningiomas. Acta Neuropathol. 1996, 91(5), 504-10.
  2. Kasuya, H.; Kubo, O.; Tanaka, M.; Amano, K.; Kato, K.; Hori, T. Clinical and radiological features related to the growth potential of meningioma. Neurosurg Rev. 2006, 29(4), 293-6; discussion 296-7.
  3. Qi, Z.Y.; Shao, C.; Huang, Y.L.; Hui, G.Z.; Zhou, Y.X.; Wang, Z. Reproductive and exogenous hormone factors in relation to risk of meningioma in women: a meta-analysis. PLoS One. 2013, 8(12), e83261.
  4. Pines, A. Hormone therapy and brain tumors. Climacteric. 2011, 14(2), 215-6.
  5. Boström, J.; Meyer-Puttlitz, B.; Wolter, M.; Blaschke, B.; Weber, R.G.; Lichter, P.; Ichimura, K.; Collins, V.P.; Reifenberger, G. Alterations of the tumor suppressor genes CDKN2A (p16(INK4a)), p14(ARF), CDKN2B (p15(INK4b)), and CDKN2C (p18(INK4c)) in atypical and anaplastic meningiomas. Am J Pathol. 2001, 159(2), 661-9.
  6. Goutagny, S.; Yang, H.W.; Zucman-Rossi, J.; Chan, J.; Dreyfuss, J.M.; Park, P.J.; Black, P.M.; Giovannini, M.; Carroll, R.S.; Kalamarides, M. Genomic profiling reveals alternative genetic pathways of meningioma malignant progression dependent on the underlying NF2 status. Clin Cancer Res. 2010,16(16), 4155-64.
  7. Sievers, P.; Hielscher, T.; Schrimpf, D.; Stichel, D.; Reuss, D.E.; Berghoff, A.S.; Neidert, M.C.; Wirsching, H.G.; Mawrin, C.; Ketter, R.; Paulus, W.; Reifenberger, G.; Lamszus, K.; Westphal, M.; Etminan, N.; Ratliff, M.; Herold-Mende, C.; Pfister, S.M.; Jones, D.T.W.; Weller, M.; Harter, P.N.; Wick, W.; Preusser, M.; von Deimling, A.; Sahm, F. CDKN2A/B homozygous deletion is associated with early recurrence in meningiomas. Acta Neuropathol. 2020, 140(3), 409-413.
  8. Mirian, C.; Duun-Henriksen, A.K.; Juratli, T.; Sahm, F.; Spiegl-Kreinecker, S.; Peyre, M.; Biczok, A.; Tonn, J.C.; Goutagny, S.; Bertero, L.; Maier, A.D.; Møller Pedersen, M.; Law, I.; Broholm, H.; Cahill, D.P.; Brastianos, P.; Poulsgaard, L.; Fugleholm, K.; Ziebell, M.; Munch, T.; Mathiesen, T. Poor prognosis associated with TERT gene alterations in meningioma is independent of the WHO classification: an individual patient data meta-analysis. J Neurol Neurosurg Psychiatry. 2020, 91(4), 378-387.
  9. Ruttledge, M.H.; Sarrazin, J.; Rangaratnam, S.; Phelan, C.M.; Twist, E.; Merel, P.; Delattre, O.; Thomas, G.; Nordenskjöld, M.; Collins, V.P.; et al. Evidence for the complete inactivation of the NF2 gene in the majority of sporadic meningiomas. Nat Genet. 1994, 6(2), 180-4.
  10. Teranishi, Y.; Okano, A.; Miyawaki, S.; Ohara, K.; Ishigami, D.; Hongo, H.; Dofuku, S.; Takami, H.; Mitsui, J.; Ikemura, M.; Komura, D.; Katoh, H.; Ushiku, T.; Ishikawa, S.; Shin, M.; Nakatomi, H.; Saito, N. Clinical significance of NF2 alteration in grade I meningiomas revisited; prognostic impact integrated with extent of resection, tumour location, and Ki-67 index. Acta Neuropathol Commun. 2022, 10(1), 76.
  11. Ko, C.C.; Zhang, Y.; Chen, J.H.; Chang, K.T.; Chen, T.Y.; Lim, S.W.; Wu, T.C.; Su, M.Y. Pre-operative MRI Radiomics for the Prediction of Progression and Recurrence in Meningiomas. Front Neurol. 2021, 12, 636235.
  12. Bhandari, A.; Marwah, R.; Smith, J.; Nguyen, D.; Bhatti, A.; Lim, C.P.; Lasocki, A. Machine learning imaging applications in the differentiation of true tumour progression from treatment-related effects in brain tumours: A systematic review and meta-analysis. J Med Imaging Radiat Oncol. 2022, 66(6), 781-797.

Reviewer 3 Report

The authors presented their own experiences against the sphenoid ridge meningioma.  They noticed the surgical results of irregular shape meningioma were poor. They separated their clinical cases into two group and evaluated each other. As the classification of two groups was subjective, the author utilized the sphericity number. The Sphericity of each group was presented in figure 2B.

Any particle which is not a sphere have sphericity less than 1. On the other hand, the sphericity of each group in the figure 2B was over 1.  2. In some cases, maybe 7 or 8 cases of the irregular group, the sphericity was 1, which means complete sphere.  Why the authors classified that sphericity 1 was irregular?  

Author Response

Dear Reviewer

Thank you for reading our manuscript and critically reviewing it, which will help us improve it to a better scientific level and make it more understandable to the readership.

In the following we would like to respond to your remarks:

The reviewer is absolutely right that the violin plot might be misleading for the readership. The created Violin Plot using GraphPad Prism 8 was an extended violin which is created as a means to show an estimated data density distribution. However, the formula of sphericity by Hakon Wadell [1] does not enable values greater than 1. Hence, an extended Violin plot using the kernel density estimation might not make a physical sense in this case [2]. We thank the reviewer for this remark. Therefore, we have revised the figure in section “3.2 Patient characteristics in regularly and irregularly shaped medial sphenoid wing meningiomas” and we newly created a box-plot (Figure 2b) displaying the median values (vertical line), mean values (+ symbol) and range (minimum-maximum displayed by whiskers). Despite the significant difference of sphericity among regularly and irregularly shaped MSWMs, a few cases were judged independently by two reviewers as irregularly shaped MSWM and the Hakon Wadell method revealed a comparatively high sphericity. Hence, a further validation of this quantitative measure is still needed in larger cohorts. We added this limitation of the present investigation to the section “4. Discussion”.

References

  1. Wadell, H.Volume, shape, and roundness of quartz particles. J Geol. 1935, 43(3), 250–280. 
  2. Thrunn, M.C.; Gehlert, T.; Ultsch, A. Analyzing the fine structure of distributions. PLoS One. 2020 Oct 14;15(10):e0238835.

Reviewer 4 Report

I thank the authors for this interesting manuscript. The scientific foundation is good. The pro and con have been thoroughly discussed. I read the manuscript with interest and delight. I only have one small question: do the authors use prophylactic administration of anti-epileptic drugs during and after meningioma surgery? Did seizures occur after surgery and could a difference between regular and irregular shaped meningiomas be found?

Author Response

Dear Reviewer

Thank you for reading our manuscript and critically reviewing it, which will help us improve it to a better scientific level and make it more understandable to the readership.

In the following we would like to respond to your remarks:

The reviewer is absolutely right that postoperative seizures are an important issue in meningioma surgery because of the known impact of epilepsy and the need of antiepileptic drug treatment on health-related quality of life (HRQOL) [1] To date, there is only retrospective data regarding the prophylactic prescription of anti-epileptic drugs (AED) in seizure-free meningioma patients who undergo surgery for meningioma. However, a recent systematic review and meta-analysis addressed this issue and reported that AED prophylaxis in meningioma patients without the history of seizures did not significantly reduce the frequency of postoperative seizures (Odds ratio: 1.26, 95% confidence interval: 0.60-2.78) compared with those who received no AED prophylaxis [2]. Hence, there is the need for prospective data investigating this issue before a recommendation supporting AED prophylaxis can be established. In our center we do not prescribe AED prophylaxis in seizure-free patients before or after surgery.

In the present series, 8 preoperatively seizure-free patients (8/74; 10.8%) had seizures in the postoperative period. Five (5/43; 11.6%) patients with a regularly shaped MSWMs had postoperative new-onset seizures, whereas 3 (3/31; 9.7%) of the irregularly  shaped MSWMs had postoperative new-onset seizures (Fisher´s exact test (two-sided): p = 0.99). Nevertheless, this potential association regarding shape and seizure might have to be evaluated in other anatomic locations (e.g., convexity, parasagittal, falcine) of meningiomas as a potential risk factor for postoperative seizures. A recent systematic review and meta-analysis revealed that non-skull base location (e.g., convexity, parasagittal, falcine)  is an independent predictor of postoperative seizures after meningioma surgery [3]. We strive to investigate the impact of shape in non-skull base meningiomas on postoperative seizures in a future investigation.

References

  1. Waagemans, M.L.; van Nieuwenhuizen, D.; Dijkstra, M.; Wumkes, M.; Dirven, C.M.; Leenstra, S.; Reijneveld, J.C.; Klein, M.; Stalpers, L.J. Long-term impact of cognitive deficits and epilepsy on quality of life in patients with low-grade meningiomas. Neurosurgery. 2011, 69(1), 72-8; discussion 78-9.
  2. Delgado-López, P.D.; Ortega-Cubero, S.; González Bernal, J.J.; Cubo-Delgado, E. Seizure prophylaxis in meningiomas: a systematic review and meta-analysis. Neurologia (Engl Ed). 2023, 38(4), 291-302.
  3. Lu, V.M.; Wahood, W.; Akinduro, O.O.; Parney, I.F.; Quinones-Hinojosa, A.; Chaichana, K.L. Four Independent Predictors of Postoperative Seizures After Meningioma Surgery: A Meta-Analysis. World Neurosurg. 2019, 130, 537-545.e3.

Reviewer 5 Report

The authors reported significance of irregularity in the tumor shape on cranial nerve dysfunction, proliferation, and progression-free survival (PFS) in medial sphenoid wing meningioma (MSWM). Irregular-shaped MSWM was associated with postoperative cranial nerve deficits, highly proliferating activity (a high value of MIB-1 labelling index) and shorter PFS. The manuscript is well constructed and includes an interesting analytic method for evaluating irregularity of the tumor surface (Sphericity), but it is well known that an irregular shape of the tumor is often associated with malignancy of MSWM. Consequently, the present manuscript does not include a novel information, but it is valuable that the authors quantitatively evaluated a shape irregularity of tumor cell. I would like to make some comments as follows.

Comments

#1 The authors described 11 patients developed new cranial nerve deficits after surgery. The authors should describe how many patients showing preoperative cranial nerve deficits deteriorated the impaired functions in both irregular-shaped and regular-shaped MSWM.

#2 Among factors that may be related to neurological deficits including postoperative dysfunction of cranial nerves and an extent of tumor resection in MSWM, tumor vascularity and tumor consistence are also very important factors. The authors should describe how these factors affected on postoperative outcome as well as other factors.

#3 Irregular-shaped MSWM often presents disappearance of arachnoid membrane over the meningioma. Some of them were resulted in tumor resection of Simpson grade IV and may make injury of normal brain and vital structures including cranial nerves and arteries. Please discuss what problems the disappearance of arachnoid membrane made in the surgery of MSWM, and whether there was a difference between irregular- and regular-shaped tumors.

#4 The authors reported higher MIB-1 labelling index was associated with irregular-shaped MSWM compared with regular-shaped MSWM. This high proliferation rate in irregular shaped MSWM causes worse survival and postoperative dysfunction of cranial nerves in the tumor. How a high proliferation of the tumor promotes an irregular shape of MSWM requires to be explained about the mechanism.

#5 Please correct errors in the text.

Page 6, line 3; which which

Page 16, line 2 from bottom; The shape of meningiomas shape

Quality of English language is almost acceptable. Partly, it is difficult to understand the meaning.  

Author Response

Dear Reviewer

Thank you for reading our manuscript and critically reviewing it, which will help us improve it to a better scientific level and make it more understandable to the readership.

In the following we would like to respond to your remarks:

  1. We absolutely agree with the reviewer that also the postoperative course of those patients with already preoperatively existing cranial nerve deficits is of importance. Hence, we added the new section “6 Association between shape and postoperative functioning of preoperatively existing cranial nerve deficits” to the results. Twenty-six patients had either a single cranial nerve deficit or multiple cranial nerve deficits of the cranial nerves II, III, IV or VI. Postoperatively, 5 of those patients (5/26;) had a postoperative worsening of the preoperatively existing cranial nerve deficits. There were three deteriorations of the visual acuity (CN II) and two patients developed a worsening of preoperatively existing oculomotor nerve palsies. One of the postoperatively further deteriorating cranial nerve palsies was observed in a regularly shaped MSWM (1/14; 7.1%), whereas 4 worsenings were observed in irregularly shaped MSWMs (4/12; 33.3%; Fisher´s exact test (two-sided): p = 0.15).
  2. The reviewer is absolutely right that the blood supply, consistency as well as the disruption of the arachnoid layer are of paramount regarding the surgical therapy. The arachnoid membrane partitions meningioma from normal brain parenchyma. Hence, this arachnoid layer is sometimes disrupted, and can make meningioma surgery challenging in critical locations with adjacent neurovascular structures [1]. Al-Mefty already described the importance of the arachnoid layer among his main principles of skull base surgery: Remove bone to minimize injury to the brain; respect the arachnoid plane, repair vessels with meticulous care, plan for reconstruction and closure with the opening, be vigilant to displace neurovascular anatomy, and strive for total removal during the first surgery as far as possible [2]. Hence, we re-evaluated our data to add this information to the present manuscript. Tumor consistency was graded as soft or firm using the Zada´s consistency grading system [3]. Soft MSWMs were defined as those amenable to be resected totally or mainly with suction (constituting Grade 1 and 2), whereas firm MSWM were defined as those requiring sharp resection, ultrasonic aspiration or with calcified portions (constituting grade 3, 4 or 5). Pial vascularization of the MSWM was evaluated using T2-weighted MR-images and peritumoral flow voids were graded as present or absent [4]. Integrity of the arachnoid layer was investigated using T2-weighted MR-images and was classified as intact in case of the presence of cerebrospinal fluid at the brain/MSWM surface [5] The proportions of patients with cystic appearance, peritumoral flow voids in T2-weighted MR-images suggesting pial blood supply, disruption of arachnoid layer, and firm tumor consistency have been added to the table 1 in section „1 Patient characteristics. Furthermore, the proportions of patients with those characteristics among regularly or irregularly shaped MSWMs have been added to the table 2 in section „3.2 Patient characteristics in regularly and irregularly shaped medial sphenoid wing meningiomas“. Moreover, the association between tumor consistency, disruption of arachnoid layer, and pial blood supply with MIB-1 index is reported in section „3.4 Association between shape and MIB-1 index“. Fourteen patients with a firm MSWMs had an increased MIB-1 index (14/24; 58.3%), and 23 patients with a firm MSWM had a normal MIB-1 Index (23/50; 46.0%), respectively (p = 0.46). Fourteen patients with a peritumoral flow void suggesting pial blood supply had a normal MIB-1 index (14/50; 28.0%), whereas 10 patients with a pial blood supply had an increased MIB-1 index (10/24; 41.7%; Fisher´s exact test (two-sided): p = 0.29). Fifteen MSWM patients (15/50; 30.0%) with a normal MIB-1 index had a disrupted arachnoid layer, and 11 MSWM patients (11/24; 45.8%) with an increased MIB-index had a disrupted arachnoid layer, respectively (p = 0.20). Additionally, in section „3.5 Association between shape and new cranial nerve morbidity after medial sphenoid wing meningioma surgery“ the impact of tumor consistency, disruption of arachnoid layer, and pial blood supply on postoperative new cranial nerve deficits is described. Firm MSWM consistency was present 5 patients (5/11; 45.5%) with new cranial nerve deficits, whereas 32 (32/63; 50.8%) patients had a firm MSWM without new postoperative cranial nerve deficits (Fisher´s exact test (two-sided): p = 0.99). Two patients (2/11; 18.2%) with a pial blood supply had new postoperative cranial nerve deficits, and 22 patients (22/63; 34.9%) with pial blood supply had no new cranial nerve deficit, respectively (p = 0.49). Disruption of the arachnoid layer was present in 6 (6/11; 54.5%) patients with new cranial nerve deficits, and disruption of the arachnoid layer was observed in 20 (20/63; 31.7%) patients without a new postoperative cranial nerve deficit, respectively (p = 0.18).
  3. We agree with the reviewer that future tailored care for meningioma patients necessitates the molecular investigation of the association between cell proliferation and meningioma shape. Therefore, we revised the section “4. Discussion”. Irregular shape is thought to result from substantial heterogeneity in proliferative activity in various subregions of the tumor, with some areas exhibiting significantly increased growth rates [6, 7]. Thus, meningioma shape has been suggested as a means to estimate the meningioma grade preoperatively [8]. Future studies will have to investigate the molecular background of regularly and irregularly shaped meningiomas to further understand the pathophysiology and develop potential tailored adjuvant therapies. Homozygous focal deletions of the cyclin-dependent kinase inhibitor 2A (CDKN2A) gene have been observed in highly proliferative meningiomas such as anaplastic WHO grade 3 meningiomas [9-11]. CDKN2A/B homozygous deletions on the chromosome 9p21 has been demonstrated to be predictive regarding progression-free survival in WHO grade 2 and 3 meningiomas [11]. However, in the series of Sievers et al. [11] analyzing 528 patients only 4.9% had a homozygous CDKN2A/B deletion. Hence, also other characteristics such as mutations of the TERT promoter will have to be investigated in irregularly shaped meningiomas to further understand the molecular pathophysiology of meningioma shape. A meta-analysis investigating data from 677 patients revealed that TERT alterations predicted progression-free survival independent of the WHO grades in meningiomas [12]. Furthermore, loss of the NF2 suppressor gene at chromosome 22 has been demonstrated in a higher percentage (50-60%) of meningiomas and supratentorial meningiomas with NF2 alterations had higher MIB-1 indices and a shorter time to tumor progression [13, 14]. Those emerging molecular markers and their association with shape will be of importance in future studies to further understand the potential mechanisms inducing intratumoral heterogeneity of the cell proliferation. We strive to investigate the association between those molecular items with meningioma shape in a future study.

References

  1. Bi, W.L.; Dunn, I.F. Current and emerging principles in surgery for meningioma. Chin Clin Oncol. 2017, 6(Suppl 1), S7
  2. Al-Mefty, O. Operative Atlas of Meningiomas. Philadelphia: Lippincott-Raven;
  3. Zada, G.; Yashar, P.; Robison, A.; Winer, J.; Khalessi, A.; Mack, W.J.; Giannotta, S.L. A proposed grading system for standardizing tumor consistency of intracranial meningiomas. Neurosurg Focus. 2013, 35(6), E1
  4. Friconnet, G.; Espindola Ala, V.H.; Janot, K.; Brinjikji, W.; Bogey, C.; Lemnos, L.; Salle, H.; Saleme, S.; Mounayer, C.; Rouchaud, A. MRI predictive score of pial vascularization of supratentorial intracranial meningioma. Eur Radiol. 2019, 29(7), 3516-3522
  5. Liu, Y.; Chotai, S.; Chen, M.; Jin, S.; Qi, S.T.; Pan, J. Preoperative radiologic classification of convexity meningioma to predict the survival and aggressive meningioma behavior. PLoS One. 2015, 10(3), e0118908
  6. Liu, H.; Zhou, J.; Li, W.; Liu, G. Comparative analysis of the magnetic resonance imaging features between anaplastic meningioma and atypical meningioma. J Craniofac Surg 2016, 27, e229-33
  7. Yao, Y.; Xu, Y.; Liu, S.; Xue, F.; Wang, B.; Qin, S.; Sun, X.; He, J. Predicting the grade of meningiomas by clinical-radiological features: A comparison of precontrast and postcontrast MRI. Front Oncol 2022, 12, 1053089.
  8. Yan, P.F.; Yan, L.; Hu, T.T.; Xiao, D.D.; Zhang, Z.; Zhao, H.Y.; Feng, J. The Potential Value of Preoperative MRI Texture and Shape Analysis in Grading Meningiomas: A Preliminary Investigation. Transl Oncol 2017, 10(4), 570-577.
  9. Boström, J.; Meyer-Puttlitz, B.; Wolter, M.; Blaschke, B.; Weber, R.G.; Lichter, P.; Ichimura, K.; Collins, V.P.; Reifenberger, G. Alterations of the tumor suppressor genes CDKN2A (p16(INK4a)), p14(ARF), CDKN2B (p15(INK4b)), and CDKN2C (p18(INK4c)) in atypical and anaplastic meningiomas. Am J Pathol. 2001, 159(2), 661-9.
  10. Goutagny, S.; Yang, H.W.; Zucman-Rossi, J.; Chan, J.; Dreyfuss, J.M.; Park, P.J.; Black, P.M.; Giovannini, M.; Carroll, R.S.; Kalamarides, M. Genomic profiling reveals alternative genetic pathways of meningioma malignant progression dependent on the underlying NF2 status. Clin Cancer Res. 2010,16(16), 4155-64.
  11. Sievers, P.; Hielscher, T.; Schrimpf, D.; Stichel, D.; Reuss, D.E.; Berghoff, A.S.; Neidert, M.C.; Wirsching, H.G.; Mawrin, C.; Ketter, R.; Paulus, W.; Reifenberger, G.; Lamszus, K.; Westphal, M.; Etminan, N.; Ratliff, M.; Herold-Mende, C.; Pfister, S.M.; Jones, D.T.W.; Weller, M.; Harter, P.N.; Wick, W.; Preusser, M.; von Deimling, A.; Sahm, F. CDKN2A/B homozygous deletion is associated with early recurrence in meningiomas. Acta Neuropathol. 2020, 140(3), 409-413.
  12. Mirian, C.; Duun-Henriksen, A.K.; Juratli, T.; Sahm, F.; Spiegl-Kreinecker, S.; Peyre, M.; Biczok, A.; Tonn, J.C.; Goutagny, S.; Bertero, L.; Maier, A.D.; Møller Pedersen, M.; Law, I.; Broholm, H.; Cahill, D.P.; Brastianos, P.; Poulsgaard, L.; Fugleholm, K.; Ziebell, M.; Munch, T.; Mathiesen, T. Poor prognosis associated with TERT gene alterations in meningioma is independent of the WHO classification: an individual patient data meta-analysis. J Neurol Neurosurg Psychiatry. 2020, 91(4), 378-387.
  13. Ruttledge, M.H.; Sarrazin, J.; Rangaratnam, S.; Phelan, C.M.; Twist, E.; Merel, P.; Delattre, O.; Thomas, G.; Nordenskjöld, M.; Collins, V.P.; et al. Evidence for the complete inactivation of the NF2 gene in the majority of sporadic meningiomas. Nat Genet. 1994, 6(2), 180-4.
  14. Teranishi, Y.; Okano, A.; Miyawaki, S.; Ohara, K.; Ishigami, D.; Hongo, H.; Dofuku, S.; Takami, H.; Mitsui, J.; Ikemura, M.; Komura, D.; Katoh, H.; Ushiku, T.; Ishikawa, S.; Shin, M.; Nakatomi, H.; Saito, N. Clinical significance of NF2 alteration in grade I meningiomas revisited; prognostic impact integrated with extent of resection, tumour location, and Ki-67 index. Acta Neuropathol Commun. 2022, 10(1), 76.

Round 2

Reviewer 3 Report

New figure 2B is appropriate.  However, I could not evaluate whether the authors use the same law data in initial figure 2B and new figure 2B.

Reviewer 5 Report

The authors exactly revised the manuscript according to my comments. Now the manuscript has been much more instructive and comprehensive.